# The Advantage of Using Immunoinformatic Tools on Vaccine Design and Development for Coronavirus

**DOI:** 10.3390/vaccines10111844

**Published:** 2022-10-31

**Authors:** Jazmín García-Machorro, Gema Lizbeth Ramírez-Salinas, Marlet Martinez-Archundia, José Correa-Basurto

**Affiliations:** 1Laboratorio de Medicina de Conservación, Escuela Superior de Medicina, Instituto Politécnico Nacional, Mexico City 11340, Mexico; 2Laboratorio de Diseño y Desarrollo de Nuevos Fármacos e Innovación Biotécnológica, Escuela Superior de Medicina, Instituto Politécnico Nacional, México City 11340, Mexico

**Keywords:** SARS-CoV-2, COVID-19, bioinformatics, immunoinformatics, epitope, vaccine

## Abstract

After the outbreak of SARS-CoV-2 by the end of 2019, the vaccine development strategies became a worldwide priority. Furthermore, the appearances of novel SARS-CoV-2 variants challenge researchers to develop new pharmacological or preventive strategies. However, vaccines still represent an efficient way to control the SARS-CoV-2 pandemic worldwide. This review describes the importance of bioinformatic and immunoinformatic tools (in silico) for guide vaccine design. In silico strategies permit the identification of epitopes (immunogenic peptides) which could be used as potential vaccines, as well as nonacarriers such as: vector viral based vaccines, RNA-based vaccines and dendrimers through immunoinformatics. Currently, nucleic acid and protein sequential as well structural analyses through bioinformatic tools allow us to get immunogenic epitopes which can induce immune response alone or in complex with nanocarriers. One of the advantages of in silico techniques is that they facilitate the identification of epitopes, while accelerating the process and helping to economize some stages of the development of safe vaccines.

## 1. Introduction

Coronaviruses (CoVs) are capable of infecting mammals and humans [1,2,3]. The severe acute respiratory syndrome coronavirus 2 (SARS-CoV-2) is a member of the family Coronaviridae. SARS-CoV-2 is enveloped, the genome is positive-sense single-stranded RNA, and directly functions as an mRNA to translate two polyproteins from the ORF1a and ORF1b region, which are cleaved by two viral proteases into sixteen non-structural proteins (nsp1-16), up to six accessory (3a, 6, 7a, 7b, 8, and 9b) and four structural proteins: spike (S), envelope (E), membrane (M), and nucleocapsid (N) [4,5]. The S-protein is a transmembrane protein which is widely exposed on the surface of the virus as homotrimers. Each monomer consists of two subunits, S1 and S2. The S1 subunit contains a receptor binding domain (RBD) which binds to the ACE2 receptor. Instead, the S-protein plays an important role in the SARS-CoV-2 infection as well as in the induction of neutralizing antibody and T cell responses associated to protective immunity [4]. For that reason, the S-protein is an interesting target for the design of vaccines. However, the other proteins, both structural and non-structural, can be a target for the design and development of a vaccine, not only preventive, but also for therapeutic purposes.

Since the outbreak of SARS-CoV-2 in 2019, many efforts have been focused on monitoring and evaluating the evolution of the virus. At the end of 2020, the appearance of variants was detected, and they have been classified into variants of interest (VOI), variants of concern (VOC) and variants under monitoring (VUMs). The VOI present genetic changes that are predicted or known to affect virus characteristics (such as transmissibility, disease severity, immune escape, diagnostic or therapeutic escape) with a risk epidemiological impact suggesting a global public health emergence. The VOC has been associated with one or more of the following changes: increase in transmissibility or detrimental change in COVID-19 epidemiology; or increase in virulence or change in clinical disease presentation; or decrease the effectiveness of public health and social measures (mask, human–human distance, etc.) or available diagnostics, vaccines, therapeutics (Table 1). The VUMS present genetic changes that are suspected to affect the virus characteristics, but the evidence of phenotypic or epidemiological impact is currently unclear; thus, it requires enhanced monitoring and repeated evaluation pending new evidence [6,7,8,9]. Such information could bring insights about the pandemic behavior, the possible re-infection of those recovered patients and the infection of those vaccinated humans [10].

The COVID-19 vaccines still represent a useful medical strategy to control the pandemic. There are diverse platforms available for vaccine development which include: mRNA-based (like mRNA1273 from Moderna and BNT162b2 from Pfizer), inactivated viruses (CoronaVac from Sino Biotech and Sinovac), and adenovirus-based (JNJ-78436735 from Johnson & Johnson, ChAdOx1 nCoV-19, Sputnik-V and Ad5-nCoV from CanSino), which, despite their different technologies, finally expose the S-protein to host immune system to induce an immune response [20,21]. However, with the appearance of S-protein mutations, the available vaccines could lose effectiveness to prevent infection and to avoid serious illness and hospitalization. Interestingly, computational evolutionary analyses clearly demonstrate a substantial ability for CoVs from different subgenera to recombine. For example, through nonhomologous recombination, CoVs can obtain accessory ORFs from core ORFs, and exchange accessory ORFs with different CoV genera, with other viruses and even with hosts. Interestingly, most of these events are observed in the vicinity of the spike ORF [22]. Additionally, the spike ORF consistently emerges as an amino acid substitution (AAS) hotspot in VOC (Alpha, Beta, Gamma, Delta and Omicron) as well as VOI (Lambda) [23].

Thus, bioinformatics, immunoinformatics and molecular modeling fields (in silico) provide valuable information from sequences and structure of S-protein for the development of vaccines under a guide procedure (Figure 1). Some of these in silico techniques are the following:

(i) Genome (RNA or DNA) or protein sequence analysis of different viruses (e.g., SARS-CoV and SARS-CoV-2) depict similarities and differences. Indeed, some mathematical models have been applied to describe virus–host interactions, whereas sequence comparison analysis have also been used in the field of virology, for example in order to track the function of nucleotide/protein sequences, to understand the epidemiology of a viral group and to understand the evolution of these viral groups, etc. [24].

(ii) Delineate the tridimensional (3D) structural differences among members of CoVs (MERS-CoV, SARS-CoV and SARS-CoV-2). This technique is applied for homologous proteins in different lineages of viruses while depicting structural conformational geometries and charges, which allow the researcher to predict and describe some potential new features of these viruses, such as infection rates, morbidity, mortality etc., instead of the structural differences between the same proteins from different lineages [8].

(iii) Identification of specific protein sites (catalytic, active, motifs and domains, etc.) that could be crucial for the viral cycle or for the receptor–antibody recognition.

(iv) Mutational analysis of the variants is helpful to predict the susceptibility to infection, or even the potential changes in efficacy of vaccines against some variants of SARS-CoV-2 [9]. The background mutation rate of SARS-CoV-2 was estimated to be 0.5 × 10^−3^–1.1 × 10^−3^ substitutions/site/year, which translates to approximately 1.3–2.8 substitution/month for the entire genome [25]. However, the S-protein particularly accumulates mutations faster than other proteins [22]. This explains, in part, the appearance of variants. The emergence of the Omicron VOC is a great example, because it accumulated an unusually high number of mutations at the S-protein (that was not observed before in any other VOC [22]. Omicron variant encodes 37 amino acid substitutions in the S-protein, 15 of which are in RBD, leading to the escape of neutralizing antibodies [26].

In addition, in silico studies are able to identify conserved regions (both sequentially and structurally) and predict possible mutations [27]. The prediction of immunogenic targets with bioinformatics tools is an essential step in the development of an effective and safe COVID-19 vaccine [28].

All the new vaccine design strategies that have been employed, such as epitope-based, viral vector and nucleic acid-based, aim to design a vaccine capable of stimulating humoral immunity (antibody responses, the target antigen must contain B-cell receptor epitopes) and cellular immunity (T-cell epitopes). Therefore, it is essential to identify epitopes that are exposed and capable of stimulating the immune system (Figure 1). [29,30,31]. Additionally, epitope-based vaccines are safer than traditional vaccines [32].

## 2. Epitope-Based Vaccines Using Bioinformatic Studies

Nowadays, there is an extensive amount of scientific knowledge regarding the recognition process of antigens by the immune system [33] which allows researchers to apply the knowledge for the design and development of vaccines, for example epitope-vaccine based on immunogenic peptides [34,35]. This is possible thanks to in silico strategies that provide promising epitopes in a short time, for example, one strategy consists in the identification of peptides of protein targets capable of binding to protein receptors located on cells of the immune system inducing immune response. These regions of the proteins showed antigenic properties capable of stimulating the immune system against proteins from viruses [36]. Then, considering these advantages and using 3D exposed peptide regions from the S-protein surface, our research group published how to get immunogenic peptides as potential peptide-vaccine [37]. Additionally, under experimental and in silico studies it became possible to identify the RBD–ACE2 surfaces making it possible to mimic ACE2 peptides [38].

Another promising strategy is the design of a multi-epitope vaccine against SARS-CoV-2 [39]. In this sense, the DeepVacPred combines immunoinformatics tools: the linear B-cell epitopes, Cytotoxic T Lymphocytes (CTL) epitopes, Helper T Lymphocytes (HTL) and neural networks. Additionally, they are able to track messenger RNA mutations to ensure vaccine coverage [40].

The peptide-vaccine-based strategy shows several advantages compared to whole-organism-based vaccines, including lack of infectious potential, safety, ease of production, low allergic and reactogenic responses and low cost for synthetic production as well as easy store and transportation without strict refrigeration [41]. The peptide-vaccine base suggests that some peptides administered alone are capable of activating immune responses [42]; however, there are peptides that are degraded by human proteases, which can be predicted in silico [43].

One strategy to avoid the peptide degradation is the use of nanocarrier molecules such as dendrimers as peptide carriers, which induce a satisfactory immune response in the rabbits’ model [44]. Additionally, dendrimers can carry an antigen (monovalent) or several antigens (polyvalent). In this way, in silico methods provide advantages in the design of polyvalent epitope-based vaccines that are expected to be effective against pathogenic strains of human coronavirus (HcoV), i.e., HCoV-OC43, HCoV-SARS, HCoV-MERS and more importantly SARS-CoV-2 [45].

### 2.1. Viral Vector-Based Vaccines

New efforts have been developed to maximize vaccine efficacy towards diverse platforms with optimized safety profiles such as recombinant viral vectored vaccines [46].

Viral vector vaccines use modified viruses (viruses that do not cause infection) as the vehicle to deliver and express their genetic information into the target host cells (using the host translational machinery) in order to express self or foreign proteins in vivo [47]. Viral vectors vaccines have been classified according to their replication capacity in non-replicating viral vector and replicating vector vaccines. In both cases, cellular and humoral immunity are induced [48].

The SARS-CoV-2 vaccines that use non-replicative viral vectors are manufactured by: AstraZeneca + University of Oxford, CanSino Biological, Gamaleya Research Institute, Janssen Pharmaceutical. The vaccines that use replicative viral vectors are manufactured by Beijing Wantai Biological Pharmacy and Israel Institute for Biological Research [48,49].

Regarding this issue, there are some computational approaches related to antigen selection, epitope prediction, adjuvant selection, and toxicology and allergenicity prediction immune response through in silico techniques [31]. These strategies allow the inclusion of nuclei acids or proteins of pathogens into non-dangerous viral particles making the immune system trigger the humoral responses for the pathogens without any risk of disease [50]. Nowadays, it is possible to simulate, under molecular dynamics simulations, the viral particles as a whole [51]. These in silico studies are now capable of simulating the viral particles as carriers of immunogenic nucleic acid [52] and peptides [53].

### 2.2. Nucleic Acid Vaccines

Nucleic acid vaccines represent a novel approach for inducing protective immunity [54]. Thus, DNA [55] and RNA vectors [56] are emerging as very promising strategies.

DNA vaccines are based on DNA plasmid constructs as a vector [57]. The injected DNA encodes an antigenic protein of interest, which will induce the activation of the humoral and cellular immune systems. DNA vaccines have advantages in production (possibility of encoding multiple antigens in a single vaccine, efficient large-scale production in bacteria and cost effectiveness) and in storage (thermal stability) [48]. The platform is promising against SARS-CoV-2, since on September 2022 there are 52 studies registered in the clinical trials (ClinicalTrials.gov (accessed on 26 October 2022)) [58].

The RNA vaccines consists of a mRNA molecule that encodes the targeted antigen, and after administration to a human cell, the immunogen sequence requires only translation to produce the antigen of interest [59]. Some RNA vaccines can self-amplify replication and translation within the host being immunogenic [30]. During these processes, it is important to consider the instability of naked RNA delivery, in addition to the size of the delivered molecules. The instability of mRNA is mainly due to the ubiquitous presence of ribonucleases that actively degrade RNA. Hence, it is protected by the addition of a 5′ cap (7-methylguanosine cap) and 3′ polyA tail, which are essential in maintaining the stability. In addition, it is important to protect RNA from degradation, with regards to this, polymer and lipid formulations have shown sufficient protection of mRNA by effectively encapsulating it [30]. For this type of vaccines, the immunoinformatic tools are very important for the rational guide to guarantee the efficacy of immune responses [60]. The in silico studies allow the selection of mRNA desirable for translation, in addition to estimating biophysical size and immunogenicity, and their folding is robust to temperature [61]. It is due to in silico studies that it is possible to analyze data from experimental results as proteome to evaluate nucleic acid vaccines [62]. Also, in silico studies allows the application of the reverse vaccinology starting for nucleic acids analyses to immunogenic peptides [63].

### 2.3. Dendrimer–Peptide Complexes

There are some immunogenic peptides that have been identified by means of bioinformatic tools capable of activating the immune system [42]. However, there are some peptides that can be degraded by proteases [64] and are not capable of crossing biological barriers [65]. Thus, making peptide–dendrimer complexes has several biological advantages to activate the immune system, as showed in a recent publication of our workgroup employing bioinformatic tools [44]. Polyamidoamine dendrimers (PAMAMs) are synthetic macromolecules of different generations according to the number of branches with interesting biomedical applications. Generation 4 polyamidoamine dendrimers (G4-PAMAMs) have several biological applications such as drug, peptide, and DNA carriers. In addition, G4-PAMAM showed low toxicity. Additionally, G4-PAMAM presents a tridimensional structure that can form polar–nonpolar inner cavities due to its internal tertiary amine and methylene groups [66]. The peptide–PAMAM complex [67] can be developed under in silico studies to determine the best complexes to be developed and assayed experimentally [44].

### 2.4. Gene Therapy Strategies

Not all vaccines are preventive; actually, there are therapeutic vaccines which can be developed by combining molecular biology and in silico strategies in order to be applied in gene therapy. Gene therapy is the treatment that changes the function of a gene or several genes, through the elimination, insertion or replacement. Such is the case of clustered regularly interspaced short palindromic repeats (CRISPR) and their associated proteins (Cas) [68], which have the ability to edit the genome.

Wang L. reported the CRISPR–Cas13a system as an antiviral tool, Cas13a is activated by a specific crRNA that targets a specific single-stranded RNA sequence (SARS-CoV-2 RNA), leading to the cleavage of the target RNA and blockage of viral protein synthesis [69].

Another method for genome insertion is liposome. Liposome is one of promising vectors to deliver a molecular cargo such as DNA, RNA or peptides for therapeutic benefit. The liposomes are taken up by the patient’s cells and release the molecular cargo into the cell nucleus. Idris et al. reported a small interfering RNA (siRNA) therapeutic against SARS-CoV-2 infection using a novel lipid nanoparticle (LNP) delivery system [70].

The bioinformatics tools allow analyzing genes from SARS-CoV-2 to understand its behavior which is related to physio-pathological effects on human patients [71]. The analysis at the genetic level will allow researchers to make decisions for prevention and efficient treatments that mitigate the health problem [72].

## 3. Discussion

The SARS-CoV-2 pandemic remains active, despite the worldwide efforts of vaccines based on multiple strategies and the approval of antivirals for treatment [73]. This may be due to factors related to the environment (damage or invasion of wildlife; social, economic and health inequality, etc.) [74], with the host (immune response, immune compromise, genetic polymorphisms, etc.) [75], with the virus (accumulation of mutants, appearance of variants with the ability to increase transmissibility and evade the immune response, etc.) [76] and, very importantly, factors related to the vaccine itself (type of vaccine used, number of doses, timing between doses, finished vaccination scheme, booster dose, heterologous prime boost, etc.) [77].

In this sense, Amoutzias, G.D et al. envisage five possible scenarios [78] which are described briefly below.

Scenario 1: Structural constraints limit any further evolution of the SARS-CoV-2 S-protein. This scenario is not very feasible and is demonstrated by the duration of the pandemic (almost 3 years), the appearance of VOCs, especially Omicron [79,80], the RNA polymerase inaccuracy [81], the infection of mammalian animals other than humans and the ability to jump the species barrier (spillover) [82,83].

Scenario 2: Point Mutations, Insertions/Deletions, and/or Intra-SARS-CoV-2 recombination. The emergence of variants is not simply the addition of single amino-acid changes [84], but several other correlated mutations also need to occur to maintain the structural integrity of the virus [85]. Furthermore, it is conceivable that a recombinant between different Omicron subvariants or between Omicron and another variant arises within the next few months, in which both variants will be circulating [78]. Although the WHO Director-General Tedros Adhanom Ghebreyesus told “We are not there yet. But the end is in sight”, [86]; as long as the virus continues to circulate in the environment, the risk of the appearance of variants is latent. Actually, it has been suggested that the SARS-CoV-2 is here to stay [87] and COVID-19 will continue but the end of the pandemic is near [88].

Scenario 3: Intratypic recombinations between SARS-CoV-2 and other Sarbecoviruses. This scenario requires zoonotic transmission; a person could pass the virus to an animal infected with another closely related CoV from the SARS-CoV-2 lineage. Thus, the co-infection allows the recombination between the two viruses [78].

Scenario 4: Intertypic recombination between SARS-CoV-2 and viruses from other beta-CoV subgenus. A structural protein region from SARS-CoV-2 passes (via modular intertypic recombination) to an animal or human CoV of another Beta-CoVs subgenus [23]. Due to the adaptation, the protein undergoes rapid divergence, and the recombined virus is totally different to parental.

Scenario 3 and 4 have been found for another virus that have caused pandemics such as influenza A H1N1/pdm2009 where there was co-infection (with H1N1 classical swine + H1N1 avian + H3N2 seasonal human + H1N1 Eurasian swine viruses) that recombined and generated a new virus. Therefore, this scenario requires that two viruses be circulating [89] at the same time and also infect a host at the same time (co-infection).

Scenario 5: Accessory ORF acquisition by non-homologous recombination of SARS-CoV-2 with other coronaviruses or even other viruses/hosts or even via de novo gene birth. A SARS-CoV-2 genome acquires (via non-homologous recombination) accessory ORFs from gene duplication, (via horizontal gene transfer) from other CoVs, viruses, or hosts, or even by de novo gene birth [78]. These evolutionary processes have already been observed in beta-genus CoVs and as a result, the molecular biology of the recombinant virus changes [90,91,92,93,94].

In that sense, the pandemic will continue its course with some of the above scenarios or even a combination of them. The in silico studies can be useful following different strategies, like the analysis of viral and human sequences of nucleic acids [95] and/or proteins [96] to identify the immunogenic epitopes capable to induce the cellular or humoral immune response.

The great challenge is to identify possible sites for mutations before they emerge allowing researchers to design vaccines targeting variants that have not yet appeared, even designing vaccines based on the highly conserved sites between variants to maintain activity despite the emergence of novel variants. To address part of this problem, the design and development of pan-coronavirus vaccines that will provide broad protective immunity against multiple coronaviruses has been proposed [97,98,99].

It is noteworthy to take into consideration that structural proteins (M, N and E) are highly conserved with a low propensity to recombine and provide structural stability [100]. Indeed, it was found experimentally that SARS-CoV-2 seropositivity correlates with broad T cell reactivity of the structural virus proteins (M and N) at 200 days after infection [101].

This objective could be reached by using diverse epitope predictors which consider immunogenic algorithms together with 3D structural information of the protein targets evaluating their exposed and conserved residues.

Additionally, the combination of bioinformatics strategies has been helpful to validate experimental in vivo models [37,44]. More recently, our research group designed possible immunogenic epitopes that target the S-protein [102]. During the process of epitope identification of the protein targets, some factors of vital importance should be considered: (a) promiscuity in which the epitope is capable of recognizing different major histocompatibility complex (MHC) molecules by stimulating the immune system; (b) conservation grade of the peptides which are important for the immune stimulation and antibody recognition process and (c) surface exposure, important for the feasibility of finding the antibodies [41,102]. As part of the peptide design, it is also quite important to predict proteasome degradation as well as exploration of binding properties to MHC under in silico studies [103].

Finally, in silico strategies are useful to identify potential epitopes that could be employed in order to develop vaccines capable of generating high affinity and protective antibodies against the pathogen in different populations and ethnic groups. Regarding this issue, it is of vital importance that vaccine development studies consider evaluating the variation of different ethnic groups; due to the myriad in MHC molecules expressed called super-types. With the pandemic, vaccine research has increased. It is an interesting and promising opportunity to research peptide vaccine based by analyzing sequence and structural bioinformatic tools (Figure 1) in order to obtain immunogenic epitopes which can induce immune response alone [42] or in complex with nanocarriers [44].

## 4. Conclusions

In conclusion, it is clear that computational or in silico techniques can speed up vaccine evaluation processes, while guiding different stages of development; however, biological evaluation is required to validate analysis from the theoretical predictions. 

## Figures and Tables

**Figure 1 vaccines-10-01844-f001:**
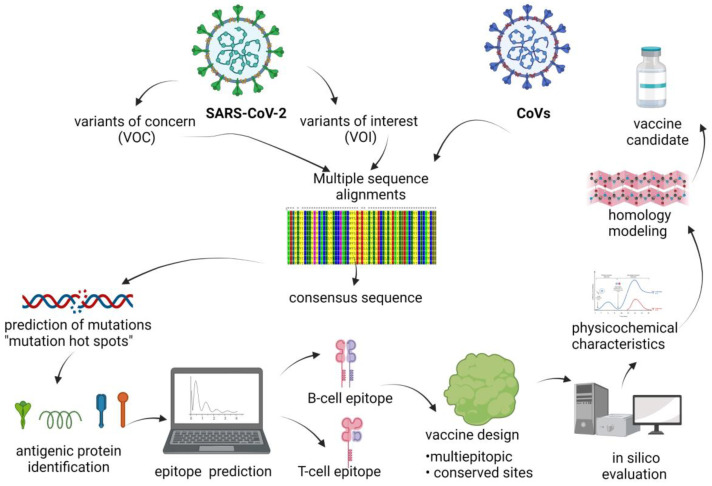
Design process of vaccine development against SARS-CoV-2. As a first step, of all the reported sequences, a consensus sequence would be obtained and analyses would be carried out to predict the probable mutation sites considering the 3D structural data using in silico studies. The data would enhance the prediction of epitopes leading to the design of an epitope vaccine. Created with BioRender.com (accessed on 29 October 2022).

**Table 1 vaccines-10-01844-t001:** Comparative characteristics between initial SARS-CoV-2 and variants of concern.

Characteristic/ Variant of Concern	Initial Virus	Alpha	Beta	Gamma	Delta	Omicron
Reproduction number (R0) or transmissibility *	2.7 [11]	4.5 [12,13,14]	≥2 [13,14]	4 [13,14]	8 [12,15]	≥10 [15]
Ability to evade immune response **	---	---	+++ [16,17]	++ [16,17]	++++ [16,17]	+++++ [18,19]
First reported	December 2019; China	14 December 2020; United Kingdom	18 December 2020; South Africa	2 January 2021; Brazil	24 March 2021; India	24 November 2021; South Africa data

* Capable of being transmitted from one person to another. ** Ability to escape the immune response, ++ low, +++ medium, ++++ high, +++++ greater ability; --- not reported. The data were taken from the indicated references.

## Data Availability

Not applicable.

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
