# Peer review of "The Advantage of Using Immunoinformatic Tools on Vaccine Design and Development for Coronavirus"

_vaccines, 2022, doi:10.3390/vaccines10111844_

Round 1

Reviewer 1 Report

General considerations:

. As it is requested on the journal's website, in the page "instructions to the authors" (https://www.mdpi.com/journal/vaccines/instructions), it is recommended that non native English speakers have their MS professionally edited to facilitate the process of peer-reviewing. This MS has a lot of gramatical issues, mostly by a confusion on the employ of the words "for" and "to". I've tried to highlight some of these errors but I still follow the journal's request and recommend the MS to be first revised by a native English speaker.

. For a MS that is produced as a review, I feel that the authors were very limited in the number of references chosen to be used as a citation on most of the sentences. Please, I ask them to include a more diverse number of references in each sentence when this is possible.

. There are some topics in the MS which doesn't add much to the work as they were constructed right now and should be further discussed to become a better source of information for the reader, especially when the text mentions other studies in the literature. For most of these topics, a lot is said about the importance of the in silico analyses on vaccine design and development. However, these topics end up being written in a very generalistic form, with very few information which is worth to read unless you go to the original citation and read the whole paper. The purpose of a review is precisely to compile the most important information of the referenced papers and save the time of the reader if he doesn't have time to read all of the literature in the area. I think this is the main issue of this MS.

 E.g:

In the topic "2.3.Dendrimer-peptide complexes", it would be an interesting addition if the authors had taken some more time to talk about their publication, cited here as reference 20. That would make this topic a lot more interesting and also it would enhance the importance of the group in this area of expertise.

In another example, in the topic "2.1. Vector viral-based vaccines", the authors have started an interesting discussion about the use of in silico studies to select antigens, predict epitopes, adjuvant selection, toxicology and allergenicity prediction. Then, all of a sudden, the topic is diverged to another complete different strategy. The MS would become so more interesting if more time was taken to have a brief discussion about how the in silico analyses could help in most of those topics, how it is done without the help of in silico analyses and why or on what extent these computational analyses have become better or more advantegeous than the classical ones.

Minor considerations:

. Page 1, line 17:  Please change "In Silico" for "in silico"

. Page 1, line 21:  Please change "In Silico" for "in silico"

. Page 1, line 23:  Please change "In Silico" for "in silico"

. Page 1, line 28:  Please change "Severe" for "severe"

. Page 1, line 32-33:  Please change "The S-protein is located on the virus surface and is widely exposed; it is a transmembrane protein as homotrimer" to "The S-protein is a transmembrane protein which is widely exposed on the surface of the virus and that is organized  in a homotrimer structure.

. Page 1, line 41: Please add a comma ( , ) in the sentence: "Since the outbreak of SARS-CoV2 in 2019 (insert a comma here) many efforts..."

. Page 2, line 55-56: Please change the sentence "However, with the appearance of S-protein mutations the avaiable vaccines can lose effectiveness for preventing infection to avoid serious ilness and hospitalization" to "However, with the appearance of S-protein mutations,  the avaiable vaccines can lose effectiveness to prevent infection and to avoid serious ilness and hospitalization."

. Page 2, line 59: Please change "In Silico" for "in silico"

. Page 2, line 58-74: This role paragraph is a lot confuse. To the reader it is not clear if next it is going to be described the guide procedure or how the in silico techniques are going to be applied. I've tried to improve understanding of the text but I ask the authors to be aware if there was any change in the sense of the text.

. Page 2, line 60: Please add the sentence "Some of these techniques are the following:" as a final sentence of this paragraph

. Page 2, line 61: Please change "a)" to "i)"

. Page 2, line 61: Please change "diverse" to "different".

. Page 2, line 62-63: Please change "...mathematical models have been applied of virus-host interaction, and currently, some other analysis have been applied in the field of virology , for example, in the analysis of viral nucleotide and protein sequences to track their function, epidemiology, and evolution." to "... not only some mathematical models have been applied to better describe virus-host interactions, but some other sequence comparison analysis have also been used in the field of virology , for example in order to track the function of nucleotide/protein sequences, to better understand the epidemiology of a viral group, to understand the evolution of these viral groups, etc."

. Page 2, line 61: Please change "b)" to "ii)"

. Page 2, line 66-68: Please change "Tridimensional (3D) structural differences among members of CoVs (MERS-CoV, SARS-CoV and SARS-CoV-2)in order to predict some possible features of these viruses, such as..." to " Delineate the tridimensional (3D) structural differences among members of CoVs (MERS-CoV, SARS-CoV and SARS-CoV-2). This technique, even when applied for homologous proteins in different lineages of viruses,  allows the researcher to predict and describe some potential new features of these viruses, such as..."

. Page 2, line 69: Please finish the sentence with a period.

. Page 2, line 75: Please change "In Silico" for "in silico".

. Page 2, line 75: Please change "possible" to "potential"

. Page 2, line 76: The link does not work anymore. Please, remove.

. Page 2, line 80-83: Please change the sentence "All the new vaccine-design strategies that have been employed such as: epitope-based, viral vector and nucleic acid-based aim to design a vaccine capable to stimulate humoral immunity..." to "All the new vaccine-design strategies that have been employed, such as epitope-based, viral vector and nucleic acid-based, aim to design a vaccine capable of stimulate humoral immunity..."

Page 2, line 80-83: Please change "safety" to "safe".

Page 2, line 86: Please change the sentence "Nowadays there is extensive amount..." to  "Nowadays there is an extensive amount..."

. Page 2, line 89: Please change "In Silico" for "in silico".

. Page 3, line 98-99: Please change "Another promising example is the design of the vaccine a multi-epitope vaccine (DeepVacPred) that combines in silico tools..." to "Another promising example is the design of a multi-epitope vaccine (DeepVacPred) that combines several in silico tools...)

. Page 3, line 109: Please change "...however, there are peptides that are degraded by human proteases, for this reason Softwares and Programs..." to "...however, there are peptides that are degraded by human proteases. For this reason, Softwares and Programs..."

.Page 3, line 124: Please change "These strategies allow to carry include..." to "These strategies allow to include..."

.Page 3, line 125: Please change "...viral particles making to immune system trigger the humoral responses..." to "...viral particles making the immune system to trigger the humoral responses..."

.Page 3, line 126: Please change "Nowadays is possible to simulate under molecular dynamics simulations the whole viral particles" to "Nowadays it is possible to simulate, under molecular dynamics simulations, the viral particles as a whole."

.Page 3, line 128: Please change "In Silico" for "in silico".

.Page 3, line 129: Please change "...of immunogenic nucleic acids peptides" to "......of immunogenic nucleic acids and peptides".

.Page 3, line 156: This topic is incomplete!

Reviewer 2 Report

Line 29: it is good practice, when mentioning SARS-CoV-2 for the first time, to include these three references (two about the genome sequence and one about the classification/naming of the virus):

doi:10.1038/s41564-020-0695-z.

doi:10.1038/s41586-020-2012-7.

doi:10.1038/s41586-020-2008-3.

Line 43-46: Please see this link from WHO and revise the sentence accordingly (with the link as reference as well):

https://www.who.int/en/activities/tracking-SARS-CoV-2-variants/

Line 46-55: It would be useful to include here the link from Carl Zimmer on the coronavirus vaccine tracker:

https://www.nytimes.com/interactive/2020/science/coronavirus-vaccine-tracker.html

Also, it would be very useful to include here the systematic review on vaccine effectiveness (DOI: 10.1093/ofid/ofac138).

Second paragraph of introduction: It would be useful to mention here that the spike region is both a recombination and a mutation hotspot in coronaviruses (see DOI: 10.3390/v14040707 and DOI: 10.1093/molbev/msab292). The background mutation rate of SARS-CoV-2 is 0.5 × 10−3–1.1 × 10−3 substitutions/site/year, which translates to approximately 1.3–2.8 substitution/month for the entire genome (see DOI: 10.1093/molbev/msac013). However, especially the Spike region accumulates mutations much faster than other regions (see DOI: 10.3390/v14040707).

The emergence of the Omicron VOC was a great example, because it accumulated an unusually high number of mutations at the spike (that was not observed before in any other VoC – see DOI: 10.3390/v14040707), that led to immune escape (see DOI: 10.1038/s41586-021-04386-2).

Several studies are attempting to circumvent this problem of the evolutionarily unstable Spike, by developing vaccines for other more conserved regions. See the thorough review of (DOI: 10.3390/v14010078) and especially sections 14 (Implications for Vaccine Design and Development) and 16 (Five Scenarios for the Future Evolution of SARS-CoV-2 during the COVID-19 Pandemic). Especially the 5 evolutionary scenarios highlight the importance of the current review (by García-Machorro et al.,) because in my opinion, we have a very long way ahead of us and need new vaccines that target other regions too.

It would be also nice, if the authors briefly discussed the pan-coronavirus vaccine approach and the approach focusing on the nucleocapsid.

Where did the authors obtain all the information regarding table 1? Please provide all the sources.

Line 146-148: the authors need to elaborate more here and not just make a brief mention. The same for line 155.

Figure 1 is rather poor.

Section 2.4 is just three lines. Something went wrong?

Line 16-21: this sentence is too long, please break it and reshape it

Line 17: this review depicts the importance of bioinformatic and immunoinformatic tools (in silico)..

Line 21: Do the authors mean sequence analysis bioinformatics tools?

Line 19 and 23: Is it nanocarriers or nonacarriers?

Line 30: 16 non-structural proteins within ORF1ab

Line 31: Please remove ORF within parentheses. It makes no sense.

Line 61-65: please rephrase.

Line 71: Please use . instead of ,

Line 84: Please rephrase.

Line 98: please rephrase.

Line 99: please italicize in silico, and elsewhere.

Line 124-127: please rephrase.

Line 136-140: please rephrase.

Reviewer 3 Report

In the present review manuscript, it is succinctly underlined the importance of bioinformatic and immunoinformatic tools, e.g., In Silico, for vaccine development on the design and/or identification of epitopes which could be used as potential vaccines as well as some nonacarriers strategies.

Although concise, the review manuscript is poorly written in general in terms of both English and scientific writing. The text should be revised by a native English- speaking professor

Here several suggestions for improving the ms

a)       “in silico” as well as other latinisms should be in italic style, e.g., lines 53, 89, 99, 202 etc

b)      Line 29 better “member of the coronaviridae family”

c)       Lin 35 better moreover instead of Then

d)      Table 1 sould be moved afteter the first time being mentioned in the text, that is line 46. The same consideration can be mate for figure 1, line 60

e)      Lines 61-74 the bulleted list should be avoided for a better reading of the text

f)        Several important references in the fieald are missing. May I suggest including https://www.nature.com/articles/s41598-022-09615-w https://www.sciencedirect.com/science/article/pii/S200103702030564X  https://pubmed.ncbi.nlm.nih.gov/34276265/ https://pubs.rsc.org/en/content/articlelanding/2022/ra/d1ra06532g https://pubs.acs.org/doi/10.1021/acs.jmedchem.1c00655

g)       Line 95 ref?

h)      Lines 102-103 empty lines between sentences should be avoided

i)        The quality of figure 1 should be improved

j)        Lines 180-181. A detailed description of the most recent immunological methods or detecting sars-cov-2 is reported here https://doi.org/10.3390/microorganisms10061193. Additional sequencing based computational approaches are also described. this reference should be included

Round 2

Reviewer 1 Report

I think the MS has been greatly improved since the previous version and should now be ready to be published

Author Response

Thank you for the feedback.

Reviewer 2 Report

The authors have extensively revised the manuscript and incorporated my suggestions/corrections. The manuscript has improved significantly is much more interesting and complete. I am happy with the content and the organization of the review. My only comments to the authors are that many sentences need to be grammatically revised. I would strongly recommend the authors to have a native English speaker help them read carefully the entire manuscript and revise it grammatically, wherever necessary. Below, I include some suggestions for language corrections, but I still strongly recommend to the authors to revise the entire manuscript for language.

Line 75: analyses

Line 78-79: are observed in the vicinity of the Spike ORF.

Line 79-81: Additionally, the Spike ORF consistently emerges as an amino  acid substitution (AAS) hotspot in VOC (Alpha, Beta, Gamma, 80 Delta and Omicron) as well as VOI (Lambda) lineages.

Line 102: was estimated to be

Line 118: of stimulating

Line 121: epitope-based vaccines are safer than traditional vaccines

Line 142: studies it became possible to

Line 170: please italicize in vivo. Is De Haan P a reference?

Line 176: please rephrase

Line 184: these in silico studies

Line 210-212: please rephrase

Line 227: simulated…developed

Line 247: Bioinformatics allows to analyze

Line 248-250: please rephrase

Line 267: infection of mammalian animals

Line 270: recombination

Line 270-271: the addition of single amino acid changes,

Line 272-274: When that review was written, both the Delta and Omicron variants were circulating, so a Delta-Omicron recombinant was possible. As we are now in October 2022, it would make more sense to revise this sentence in something like this: “Furthermore, it is conceivable that a recombinant between different Omicron subvariants or between Omicron and another variant arises within the next few months, in which both variants will be circulating [77]”.

Line 277: it has been suggested that

Line 285: beta-CoV subgenus

Line 291: and recombine and generate a new virus

Line 294: please italicize de novo

Line 298: and as a result, the molecular biology of the recombinant virus changes

Line 335-338: please rephrase

Reviewer 3 Report

accepted in the presnt form

Author Response

Thank you for the feedback.